# Porcine Digestible Peptides (PDP) in Weanling Diets Regulates the Expression of Genes Involved in Gut Barrier Function, Immune Response and Nutrient Transport in Nursery Pigs

**DOI:** 10.3390/ani10122368

**Published:** 2020-12-10

**Authors:** Francesc González-Solé, Lourdes Criado-Mesas, Carmen Villodre, Wellington C. García, Mercè Farré, Elisabet Borda, Francisco J. Pérez-Cano, Josep M. Folch, David Solà-Oriol, José F. Pérez

**Affiliations:** 1Department of Animal and Food Sciences, Animal Nutrition and Welfare Service, Autonomous University of Barcelona, 08193 Bellaterra, Spain; francesc.gonzalez.sole@uab.cat (F.G.-S.); carmenvitu@gmail.com (C.V.); wncg_100583@hotmail.com (W.C.G.); JosepMaria.Folch@uab.cat (J.M.F.); JoseFrancisco.Perez@uab.cat (J.F.P.); 2Plant and Animal Genomics, Centre for Research in Agricultural Genomics (CRAG), CSIC-IRTA-UAB-UB, 08193 Bellaterra, Spain; lourdes.criado@cragenomica.es; 3Department of Animal Production, Agrarian University of Ecuador, Guayaquil 090108, Ecuador; 4Department of Mathematics, Area of Statistics and Operations Research, Autonomous University of Barcelona, 08193 Bellaterra, Spain; farre@mat.uab.cat; 5R&D Animal Nutrition Department, Bioiberica S.A.U., 08389 Palafolls, Spain; eborda@bioiberica.com; 6Physiology Section, Department of Biochemistry and Physiology, Faculty of Pharmacy and Food Science, University of Barcelona (UB), 08028 Barcelona, Spain; franciscoperez@ub.edu; 7Nutrition and Food Safety Research Institute (INSA-UB), 08921 Santa Coloma de Gramenet, Spain

**Keywords:** gene expression, high throughput RT-qPCR, intestinal function, piglet, porcine digestible peptides

## Abstract

**Simple Summary:**

Porcine digestive peptides (PDP) are a coproduct of the heparin industry obtained from the enzymatic hydrolysis of porcine intestinal mucosa. They have proven to be a valid substitute for other high quality dietary protein sources for piglets, like spray-dried plasma (SDP), but knowledge about their influence on intestinal function is still scarce. This study found that substituting soybean ingredients with PDP and SDP to the diets of weaned piglets increased growth rate at 14 d post-weaning. In addition, the combination of PDP with SDP increased the expression of certain genes related to intestinal function in the jejunum, which suggests that this combination might have functional properties that contribute to improving the intestinal health of the pigs, although more research is needed to confirm it.

**Abstract:**

This study was conducted to investigate the effects of dietary supplementation of porcine digestible peptides (PDP), spray-dried plasma (SDP), or a combination of both, on growth performance and the expression of genes related to intestinal function of weaned pigs. A total of 180 piglets (trial 1) and 198 piglets (trial 2) were used to evaluate the partial substitution of soybean ingredients with 2% SDP or 2% PDP (trial 1), and with 3% SDP or the combination of 1% SDP and 2% PDP (SDP-PDP; trial 2) during the pre-starter period (0–14 days). The gene expression of 56 genes was quantified in a qPCR platform in jejunum and ileum samples obtained from piglets 14 d after weaning (trial 2). Piglets fed SDP, PDP and SDP-PDP had a higher body weight (BW), average daily gain (ADG) and feed efficiency (G:F) than the soybean control on day 14 (*p* < 0.05). In addition, the combination of SDP and PDP upregulated ten genes in jejunum samples (*p* < 0.05) related to intestinal function. More research is needed to confirm that gene expression upregulation by PDP in combination with SDP has an impact on intestinal function and to elucidate its underlying mechanisms.

## 1. Introduction

Weaning is a stressful period for piglets as they have to deal with a change from sow milk to a less digestible, plant-based dry solid diet that contains many ingredients that the pig has not eaten before [1]. One major consequence of weaning is a reduction in feed intake, which in turn causes a reduction in the villi height in the small intestine [2] and a disruption of the gut microbiota ecosystem with a loss of diversity [3]. After weaning, piglets are more susceptible to gut inflammatory problems and their intestinal function can be affected. This often leads to post-weaning diarrhea and increased mucosal permeability [4]. A strategy for helping piglets to reduce intestinal disturbances during this period is to reduce the inclusion of less-digestible vegetal protein sources in their feed, like soybean ingredients, and substitute them with high-quality digestible animal protein sources. Certain protein sources, like spray-dried plasma (SDP) contain biologically active components that give them physiological or regulatory functions beyond their nutritional value [5]. In particular, SDP has shown potential for modulating the immune response, reducing intestinal mucosa inflammation and maintaining its integrity [6,7,8].

Porcine digestible peptides (PDP) are a coproduct of the heparin industry and are obtained from the enzymatic hydrolysis of porcine intestinal mucosa. Currently, PDP can be used in postweaning diets as the cheapest alternative for SDP, fish meal and other sources of high-quality protein in terms of preference and digestibility [9,10]. The ability of PDP to increase villus height suggests that they may improve nutrient uptake [11]. However, there is no available literature exploring possible effects of PDP on intestinal function when replacing major soy protein ingredients in the diet. In the present study, in an initial trial we hypothesize that including PDP and SDP in a high crude protein (CP) diet partially substituting a high content of soybean ingredients might improve piglet performance after weaning (trial 1). In a second trial we also hypothesize that partially substituting soybean meal (SBM) in weanling diets with SDP or a combination of SDP and PDP could improve piglet performance and the expression of genes related to intestinal function (trial 2).

## 2. Materials and Methods 

The experimental procedures used in the two trials were approved by the Ethical Committee on Animal Experimentation of the Autonomous University of Barcelona (CEAAH 3817), and are in full compliance with national legislation following the EU-Directive 2010/63/EU for the protection of animals used for scientific purposes.

### 2.1. Animals, Housing and Diet

#### 2.1.1. Trial 1

The first trial was conducted in the weanling unit of a commercial farm throughout the pre-starter period (0–14 d post-weaning). A total of 180 male and female weaned commercial piglets ((Landrace × Large White) × Piétrain, weaned at 28 d) with a body weight (BW) of 7.5 ± SD 1.15 kg were moved to the nursery unit. These animals were not given previous access to creep feed during lactation. Piglets were distributed into two blocks according to initial BW (heavy piglets: 8.6 ± SD 0.03 kg; light piglets: 6.4 ± SD 0.02 kg). Each block contained 9 pens of 10 animals to which the three experimental treatments were randomly assigned (6 pens or replicates/treatment). Each pen (3.2 m^2^ in floor area) had a commercial non-lidded hopper (TR5, Rotecna, Agramunt, Spain) and a nipple waterer to ensure ad libitum feeding and free water access.

Treatments consisted of three different iso-protein pre-starter diets: a control diet (CON) with a high content of soybean ingredients and two extra diets with partial replacement of the extruded soybeans by 2% SDP (AP 820 P, APC Europe S.L., Granollers, Spain) or 2% PDP (Palbio 62SP, Bioiberica S.A.U., Palafolls, Spain). The basal pre-starter diet was formulated to contain 2470 kcal of net energy (NE)/kg, 19.5% CP/kg and 1.28% digestible Lys (Table 1) to meet the requirements for maintenance and growth of newly weaned piglets [12]. Diets were presented in mash form and were fed ad libitum for fourteen consecutive days. No antimicrobials or ZnO were used in the experimental diets.

#### 2.1.2. Trial 2

The second trial was conducted in a different commercial farm throughout the pre-starter (0–14 d post-weaning) and starter period (14–35 d post-weaning). A total of 198 male and female weaned commercial crossing piglets ((Landrace × Large White) × Piétrain, weaned at 21 d) with a BW of 5.7 ± SD 0.60 kg were moved to the nursery unit without transport to be used in the trial. Piglets had access to creep feed during lactation. Animals were distributed into two blocks according to initial BW (heavy piglets: 6.3 ± SD 0.02 kg; light piglets: 5.1 ± SD 0.01 kg) and each block contained 9 pens of 11 animals to which three experimental treatments were randomly assigned (6 replicates for each treatment). Each pen (3 m^2^ in floor area) had a commercial non-lidded hopper (TR5, Rotecna) and a nipple waterer to ensure ad libitum feeding and free water access. The different characteristics of this farm compared to the farm of the first trial forced some changes to the experimental design of the experiment.

Treatments consisted of three different iso-protein pre-starter diets: a CON diet with a high content of SBM and two extra diets with partial replacement of the SBM by 3% spray-dried plasma (SDP; AP 820 P, APC Europe S.L.) or a combination of 1% spray-dried plasma and 2% porcine digestible peptides (SDP-PDP; Palbio 62SP, Bioiberica S.A.U.). The basal pre-starter diet was formulated to contain 2470 kcal NE/kg, 20.5% CP/kg and 1.35% digestible Lys (Table 2) and to meet the requirements for maintenance and growth of newly weaned piglets [13]. The pre-starter diets were fed ad libitum for fourteen consecutive days and a common starter diet was also fed ad libitum from 15 to 35 d post-weaning (Table 2). No antibiotics, alternative antimicrobials or ZnO were included in the diets.

### 2.2. Data and Sample Collection

Piglets were ear tagged and individually weighed at weaning (0 d) and 14 d post weaning in trial 1 and at weaning (0 d), 7 d, 14 d and 35 d post-weaning in trial 2. Feed disappearance from each hopper was measured throughout the experimental period. The average daily feed intake (ADFI), average daily gain (ADG) and feed efficiency (G:F) were calculated for the experimental period.

One piglet per pen was sedated with a combination of zolazepam/tiletamine and xylazine, and was euthanized with a pentobarbital injection on day 14 in trial 2. Portions of 0.5 cm of jejunum and ileum tissues were collected at 30 cm and 1.3 cm from the ileo-cecal valve respectively. Intestinal sections were rinsed in phosphate-buffered saline (PBS) and immediately snap frozen in 1 mL of RNAlater (Deltalab, Rubí, Spain). Samples were stored at −80 °C until analysis.

### 2.3. Proximate Analysis of Diets

Diet proximate analyses from both trials were performed following the Association of Official Agricultural Chemists methodology: dry matter (AOAC 934.01 [14]), ash (AOAC 942.05 [15]), ether extract (AOAC 2003.05 [15]) and crude protein (AOAC 968.06 [16]). Neutral-detergent fiber was determined according to the method of Van Soest et al. [17].

### 2.4. Gene Expression Study by qPCR

Gene expression was quantified by RT-qPCR to study the expression of 56 genes in two intestinal tissues in an Open Array Real-Time PCR Platform (Applied Biosystems, Waltham, MA, USA) by the Servei Veterinari de Genètica Molecular at the Veterinary Faculty of the Universitat Autònoma de Barcelona (Spain).

The pre-amplified product was diluted 1:10 with 0.1× Tris-EDTA pH 8.0 and 6 µL was transferred to 384-well plates. These were analyzed in duplicate in Taqman Open Array gene expression plates custom-designed in a QuantStudio 12K Flex Real-Time PCR system (ThermoFisher Scientific, Waltham, MA, USA). One sample was used as an inter-plate control to check the replication of results from different plates.

### 2.5. Open Array Design

A list of 56 genes related to intestinal health were selected according to the bibliography [18,19,20,21,22,23,24,25,26,27,28,29,30,31] and included: (1) genes participating in the barrier function (*OCLN*, *ZO1*, *CLDN1*, *CLDN4*, *CLDN15*, *MUC2*, *MUC13* and *TFF3*); (2) genes that play an important role in the immune response, such as pattern recognition receptors, cytokines, chemokines and stress proteins (*TLR2*, *TLR4*, *IL1B*, *IL6*, *IL8*, *IL10*, *IL17A*, *IL22*, *IFNG*, *TNF*, *TGFB1*, *CCL20*, *CXCL2*, *IFNGR1*, *HSPB1*, *HSPA4*, *REG3G*, *PPARGC1A*, *FAXDC2* and *GBP1*); (3) genes coding for enzymes and hormones implicated in the digestion process (*GPX2*, *SOD2*, *ALPI*, *SI*, *DAO1*, *HNMT*, *APN*, *IDO1*, *GCG*, *CCK*, *IGF1R* and *PYY*); (4) genes involved in nutrient transport (*SLC5A1*, *SLC16A1*, *SLC7A8*, *SLC15A1*, *SLC13A1*, *SLC11A2*, *MT1A*, *SLC30A1* and *SLC39A4*); (5) genes involved in stress response (*CRHR1*, *NR3C1* and *HSD11B1*); and (6) four reference genes (*ACTB*, *B2M*, *GAPDH* and *TBP*).

Primers were designed spanning exon-exon boundaries or at different exons using PrimerExpress 2.0 software (Applied Biosystems) for 55 genes. The *IL8* gene primer was pre-designed by the company due to its complexity (Table A1). Possible residual genomic DNA amplification and primer dimer formation were controlled. Finally, a customized open array panel containing 56 genes was obtained.

### 2.6. RNA Extraction and cDNA Preparation

Total RNA was obtained from 100 mg of frozen intestinal tissues with the RiboPure kit (Ambion, Austin, TX, USA) following the manufacturer’s protocol. RNA concentration and purity was calculated with a NanoDrop ND-1000 spectrophotometer (NanoDrop products, Wilmington, DE, USA). RNA integrity was checked with Agilent Bioanalyzer-2100 equipment (Agilent Technologies, Santa Clara, CA, USA) following the producer’s protocol. One microgram of total RNA was reverse-transcribed into cDNA in a final volume of 20 µL. The High-Capacity cDNA Reverse Transcription kit (Applied Biosystems) and random hexamer primers were used, and the following thermal profile was applied: 25 °C, 10 min; 37 °C, 120 min; 85 °C, 5 s; 4 °C hold. A total of 25 ng of cDNA sample was pre-amplified using a 2× TaqMan PreAmp Master Mix and a 0.2× Pooled Taqman Gene Expression Custom Assays in a final volume of 10 µL. The thermal cycling conditions for the pre-amplification reactions were 10 min at 95 °C; 14 cycles of 15 s at 95 °C and 4 min at 60 °C; and a final step of 10 min at 99 °C. The pre-amplified cDNA product was stored at −20 °C until use.

### 2.7. Gene Expression Data Analysis

Gene expression data were collected and analyzed using the ThermoFisher Cloud software 1.0 (Applied Biosystems) applying the 2^−ΔΔCt^ method for relative quantification and using the sample with the lowest expression as a calibrator. Some parameters were adjusted: the maximum cycle relative threshold allowed was 26, amplification score < 1.240, quantification cycle confidence > 0.8 and the maximum standard deviation allowed between duplicates was set at <0.38. Samples that did not fit these requirements or had an inconclusive amplification status were deleted. Relative quantification values were checked for normalization by a log_10_ transformation, and all the statistical analyses were performed with R 3.4.3 software [32] and Bioconductor [33]. We carried out a one-way ANOVA and calculated the Benjamini–Hochberg false discovery rate (FDR *q*-value) to control multiple *p*-values [34], setting an upper bound for the expected proportion of false significant tests, that is, false significant treatment differences in mean expression levels between treatments. Pairwise post hoc treatment comparisons were carried out using Tukey’s honest significant difference test [35]. Statistical differences between results for the treatments were identified at *p*-values and *q*-values under 0.05 for the ANOVA and Tukey’s analysis and for the FDR, respectively.

A principal component analysis (PCA) was performed with samples as cases and gene log_10_-expressions as variables. The function PCA of the FactoMiner R-library [36] was used for dimension reduction and visualization in the first two principal dimensions. The variables factor map was restricted to genes showing cos2-qualities over 0.45 and significant differences. Finally, the heatmap visualization method was used to obtain double clustering both for genes (with a correlation-based distance:(1)d=(1−r)/2,
where *d* is the distance and *r* is the correlation coefficient, and the complete linkage hierarchical clustering) and for samples (with Euclidean distance and Ward’s D2 method). These methods were chosen based on the following: the Euclidian distance between two samples adds all the squared differences in the log-expression level of them in each gene, and then the Ward’s D2 linkage method uses the Euclidean squared-distance to cluster in a way that minimizes the increment of the variance into the resulting clusters. Correlation based distance is preferred for obtaining the gene clusters because the expression levels in different genes may not be comparable (see Murtagh and Legendre [37] for Ward’s method and Everitt [38] for clustering methods). The function heatmap.2 in the R-library gplots was used [39].

### 2.8. Performance Data Statistical Analysis

Production performance data were analyzed with ANOVA using the generalized linear model procedure of the statistical package SAS (version 9.4, SAS Inst. Inc., Cary, NC, USA). Normality and homoscedasticity were checked with Shapiro–Wilk test using the univariate procedure and Levene’s test using the generalized linear model procedure, respectively. Data were analyzed taking the experimental treatment and the block of weight as the main factors. Their corresponding interaction was also included in the model. The statistical unit was the pen of 10 pigs in trial 1 and the pen of 11 pigs in trial 2. The results are presented as least square means taking into account the Tukey adjustment. The level of significance considered was α = 0.05.

## 3. Results

### 3.1. Growth Performance

The productive performance of piglets during the first two weeks after weaning (trial 1) is summarized in Table 3. Animals fed SDP and PDP showed higher BW, ADG and G:F (*p* < 0.05) than piglets in the control group.

For trial 2, the productive performance results are shown in Table 4. Piglets of the SDP and SDP-PDP group showed greater BW, ADG and G:F than CON piglets at 14 d (*p* < 0.05). Although a numeric difference in BW was observed at 35 d between the supplemented animals and the CON group, no significant differences in BW, ADG or G:F were observed.

### 3.2. Intestinal Mucosa Gene Expression Values

After performing qPCR of the intestinal tissue, 37 genes were detected as expressed in jejunum tissue and 27 genes in ileum tissue from the 54 target genes initially included in the open array panel. The conservation of one tissue sample from the jejunum and seven from the ileum was affected and amplification was not possible. Gene expression results of the ileum tissue were not considered due to the high amount of lost samples of this intestinal section.

The results of the analysis of gene expression in the jejunum intestinal tissue (trial 2) are shown in Table 5. Although only two genes showed statistical differences between groups (*p* < 0.05, *q* < 0.05), the other 8 genes showed a trend of being modified by the diets (*p* < 0.05, *q* < 0.2). The *CLDN15* and *TFF3* genes from the barrier function group and the *SLC11A2*/*DMT1* gene from the nutrient transport group showed a tendency to higher expression for the SDP-PDP treatment compared to the CON treatment (*p* < 0.05). From the immune response functional group, the SDP-PDP group showed a trend of having higher expression levels in three genes: *GBP1* compared to the SDP group, *IFNGR1* compared to the CON group, and *TLR4* compared to the other two groups (*p* < 0.05). Finally, the SDP-PDP group had a higher expression level than the CON and SDP groups in two enzyme-coding genes (*HNMT* and *APN*) that was only significant for *HNMT* (*q* < 0.05). The gene coding for a protein that metabolizes oxidation products, *SOD2*, also showed a significant increase when comparing the SDP-PDP group with the CON and SDP groups.

A PCA was carried out to determine the correlation in the gene expression values among samples distributed in the three diet groups (Figure 1). The sample identifier numbers are represented in the individual factor map (a) and are colored according to the diet assigned to each sample. The variables factor map (b) shows which genes are correlated along the samples. The 2D representation preserves 60% of the total variance (61.27). The results show a correlation in the gene expression pattern. In this tissue, eight of the nine significant genes (*APN*, *HNMT*, *SLC11A2*, *CLCN15*, *IFNGR1*, *TLR4*, *GBP1* and *SOD2*) are well represented in 2D (long arrows) and all of them fall in the first quadrant. Furthermore, the blue colored samples in the SDP-PDP group fall in the same quadrant or near to it, which indicates medium to high expression levels. Therefore, these plots clearly show a relationship between the highest expression levels in these significant genes and the SDP-PDP diet group. The CON and SDP groups are mixed and do not show high expression values.

A heatmap was also made in order to observe overall similarities among the gene expression profiles of the animals (Figure 2). On the heatmaps, the samples from the SDP-PDP treatment tend to group in pairs with others in the same group in the first step (the shortest branches). In addition, most SDP-PDP samples fall on one side of the dendrogram, the side with the highest number of red pixels in many genes, indicating a higher level of expression of these genes within the SDP-PDP group. Clustering of genes does not show any evidence of an association by functional group.

## 4. Discussion

This study found that substituting soybean ingredients with either 2–3% PDP, SDP or both to the diets of weaned piglets increased growth rate early after weaning. There are two main reasons that could be suggested and contribute to the observed results, such as the reduction of the negative impact of the soybean ingredients and the beneficial effects of the PDP and SDP. Unfortunately, our experimental design does not allow discrimination between them. The diets in the present study were formulated to contain a high inclusion of soybean ingredients, such as soybean meal 44% or 47% CP, extruded soybeans and soybean protein concentrate 56% CP. The β-sheet structures present in the secondary structure of raw legume proteins and the intermolecular β-sheet aggregates derived from heating are negatively correlated with feed digestibility values [40]. The β-sheet structures represent more than 30% of the secondary structure of soybean seeds, while they represent less than 10% in the ingredients of animal origin [41]. Consequently, the flow of protein into the distal parts of the gastrointestinal tract tends to be faster when more soybean ingredients are included, which promotes protein fermentation and selective growth of proteolytic bacteria [41]. Furthermore, soybeans contain anti-nutritional factors, such as antigens, trypsin inhibitors and lectins, which can produce digestive disorders and reduce nutrient availability [42,43]. The consequences of high inclusion of soybean ingredients, aggravated by the situation of post-weaning anorexia, can lead to intestinal disorders, such as post-weaning diarrhea [2,44]. This can negatively affect the animal growth performance. In the present study, the reduction of soy ingredients accounts for 3–5% of the diet, being replaced by either SDP, PDP, or both. In contrast, proteins of animal origin as well as animal protein hydrolysates have higher palatability than vegetal proteins [10,45,46], which can translate into an increased feed intake after weaning.

The substitution of soybean ingredients with SDP improved the productive performance at the end of pre-starter period of both trials. The results reported herein are similar to other studies that showed that SDP improved performance, especially when piglets were challenged with experimental infection and did not receive in-feed medication [47]. Spray-dried plasma is commonly used because it can stimulate feed intake due to its palatability [48], as observed in trial 1. Its beneficial effects are explained by the preservation of blood immunoglobulins, growth factors and bioactive peptides or compounds during the spray-drying process. These components can interact with the gut-associated lymphoid tissue [49], therefore preserving the small intestinal barrier function and reducing intestinal inflammation and damage [6].

Consumption of feed containing the hydrolysates, PDP or SDP-PDP, enhanced the productive performance of piglets compared to their corresponding control group at the end of the pre-starter period, and these groups were equivalent to the SDP group. This result is in accordance with other authors who also obtained similar growth performance by feeding PDP and SDP in early-weaned piglets [50]. Some research data have shown improved growth performance, feed intake and efficiency of animals fed PDP compared to other high-quality protein sources like fish meal [45]. Other studies found that the inclusion of PDP in weanling diets improved villus height of the small intestine compared to some sources of intact protein like SDP [51] or fish meal [11], which can be considered a good indicator of nutrient uptake. Part of the beneficial effects of PDP on growth performance could be due to its content of short-chain peptides that are more easily absorbed by pigs than intact proteins [52] or even free amino acids [53].

This is the first study performed using gene expression to provide some inputs about the effects of PDP on intestinal function. The outcome of the PCA individual factor map of the gene expression shows that the factorial scores of the SDP-PDP group tended to be closer to each other and more separate from the CON and SDP group scores. In addition, the correspondence of most of the arrows representing the significant genes in the PCA variables map and most of the samples in the SDP-PDP group in a similar position indicates that there is a relationship between the higher expression levels in these genes and the differences that the SDP-PDP group showed from the CON and SDP groups. Heatmap representation also helped to visualize a partial clustering of the samples from the SDP-PDP treatment in the jejunum and ileum. In line with this, the statistical ANOVA and Tukey’s test showed a stronger effect of the SDP-PDP diet than SDP with respect to the CON diet. Although all results agree, ANOVA and Tukey’s test deal with mean treatment values while heatmaps and PCA-plots reflect individual performance. The combination of inferential treatment comparison techniques (ANOVA and Tukey, essentially) and powerful exploratory methods (PCA and heatmap) provide a clearer and dual (individuals and genes) idea of the differences in gene expression.

Proteins coded by the genes analyzed in the current study participate in the barrier function of gut cells, in nutrient transport in the mucosa, in digestion, in the immune response and in the metabolization of oxidation products.

Some trends in the jejunum gene expression due to diets may indicate a potentiation of the epithelial structure by SDP-PDP, because, for example, the TFF3 gene participates in epithelial restitution and maintenance of intestinal mucosa integrity [24]. However, the CLDN15 gene, which expression was also modified, codes for a pore-forming protein [22], which is important for the normal-sized morphogenesis of the small intestine [25].

Diets also had an effect on the expression in the jejunum of genes related to the immune response and metabolization of oxidation products. Again, the SDP-PDP diet showed more changes than the SDP diet alone. Expression of immune response genes TLR4, IFNGR1 and GBP1 showed a trend to increase in the SDP-PDP group, TLR4 compared to the CON and SDP groups; IFNGR1 compared to the CON group and GBP1 compared to the SDP group.

TLR4 is a receptor involved in the recognition of lipopolysaccharide, a major cell wall component of Gram-negative bacteria [19], and IFNGR1 is part of the receptor that mediates the biological effects of IFN-γ [54]. The TLR4 and IFNGR1 genes have been reported to be upregulated in animals under stress and with infectious conditions in order to activate the innate immune response and fight against pathogens properly [24,28,55]. Thus, the tendency for upregulation of TLR4 and IFNGR1 might suggest that piglets fed with SDP-PDP seem to be more prepared for controlling infective processes and other intestinal challenges that can occur during the weaning period. In addition, GBP1 is a GTPase that regulates the inhibition of proliferation and invasion of endothelial cells. It protects against epithelial apoptosis induced by inflammatory cytokines and subsequent loss of the barrier function [27]. Upregulation of GBP1 by the SDP-PDP diet, although without being statistically significant, is probably related to IFNGR1 upregulation because GBP1 expression is strongly induced by IFN-γ [27], although an upregulation of IFN-γ was not observed in this study.

Focusing on nutrient transport, only an up-regulatory trend in the expression of the SLC11A2/DMT1 gene was found in animals fed with the SDP-PDP combination compared to the CON group. The divalent metal transporter (SCL11A2/DMT1) is located on the apical surface of the enterocyte and is involved in the intestinal Fe uptake [18]. SLC11A2/DMT1 gene expression is upregulated in circumstances of low Fe intake [26] and hyperglycemia conditions [56]. We have no evidence of differences in the Fe content or glycemic levels among experimental diets; therefore, the reason why the SDP-PDP treatment influenced the expression of SLC11A2/DMT1 should be researched further.

The enzyme-coding gene HNMT was upregulated and the gene APN showed a trend to increase due to the SDP-PDP diet. Kröger et al. [31] reported that high dietary inclusion of fermentable CP increased the HNMT expression in the colon, which is a histamine-degrading enzyme. They determined that the histamine catabolism activity of HNMT counter-regulated the increased production of this biogenic amine, reducing the fecal score of the piglets fed with a high fermentable CP diet. Considering that all diets had an elevated CP level, an increase in this enzyme could show that SDP-PDP was attenuating the inflammatory effects of histamine more efficiently than other groups. On the other hand, APN is a Zn-dependent enzyme that takes part in the final digestion of peptides [56]. Its upregulation in the small intestine has been documented with products considered beneficial for intestinal health, such as the probiotic *Lactococcus lactis* [23]. Therefore, its increase due to the SDP-PDP diet may also be considered a positive physiological change.

Regarding the antioxidant defense mechanisms, we observed here that the expression of the SOD2 gene was also increased in the SDP-PDP group compared to both the SDP and CON groups. This mitochondrial enzyme is considered the first defense against reactive oxygen species (ROS) formed during normal cell metabolism [30]. Elimination of ROS by SOD2 can be considered as an anti-inflammatory effect due to the important role that ROS plays in triggering and promoting inflammation [57]. As well as the TLR4 gene, expression of SOD2 is stimulated by lipopolysaccharides but cytokines or ROS can also upregulate it [58]. Thus, changes in SOD2 gene expression can be derived from the TLR4 upregulation also induced by the SDP-PDP intervention.

The gene expression results of this trial might indicate that the combination of PDP and SDP had some effects on intestinal function, although most of the differences found among treatments only could be considered as a trend (*p* < 0.05, *q* < 0.2). First of all, the tendency for upregulation of TFF3 and CLDN15 could be considered a potentiation of the epithelial structure of the gut, which would be related to the increase of villi height observed in previous studies with PDP [51]. Secondly, we could speculate that animals fed PDP-SDP might be more prepared for controlling infective processes and defending against other hazardous processes due to the up-regulatory trend in the expression of the immune response genes (TLR4, IFNGR1 and GBP1) and the upregulation of genes related to the degradation of toxic products of metabolism (SOD2 and HNMT). In addition, the trend to increase the expression of the APN gene could suggest an improvement in the digestion of protein. The underlying mechanisms that produced these effects are still unknown and the literature exploring possible functional effects of PDP is still scarce. More research is needed to confirm that gene expression upregulation by PDP in combination with SDP has an impact on intestinal function and to elucidate the underlying mechanisms that are responsible of these effects. Furthermore, it might be interesting to investigate if the effects on gene expression observed in this study are only attributable to the addition of PDP or if it is a synergy of PDP with SDP that is producing the effects, as SDP alone did not show them.

## 5. Conclusions

The present study suggests that substituting soybean products with animal protein sources like PDP or SDP increases growth performance of weanling piglets at the end of the pre-starter period. In addition, it indicates that PDP can substitute or complement SDP because it showed the same effect on the growth performance of the piglets during this period. Furthermore, changes in the gene expression of the jejunum produced by the SDP-PDP diet suggest that this treatment might be able to produce beneficial effects in the epithelial structure of the gut, in the defensive capacity of the intestine against threats and the digestion of proteins. More research is needed to confirm that gene expression upregulation by PDP in combination with SDP has an impact on intestinal function and to elucidate the underlying mechanisms that are responsible for these effects.

## Figures and Tables

**Figure 1 animals-10-02368-f001:**
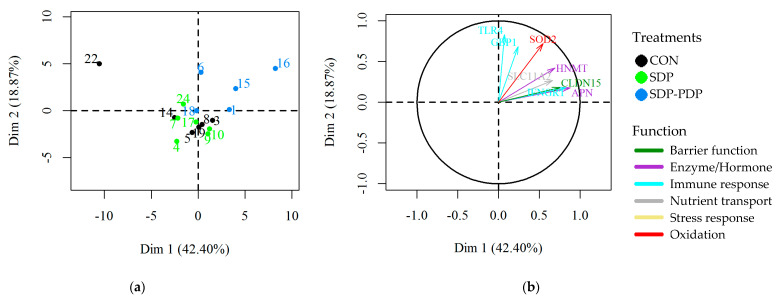
Principal component analysis (PCA): (**a**) Samples’ picture from jejunum (individuals factor map); (**b**) gene expressions arrow diagram from jejunum (variables factor map); CON: control diet; SDP: diet with 3% spray-dried plasma inclusion; SDP-PDP: diet supplemented with a combination of 1% spray-dried plasma and 2% porcine digestible peptides. Samples are labeled with different colors depending on the treatment, and the gene expressions arrows diagram on the right. Only those genes with *p* < 0.05 and having a long arrow have been depicted. Each arrow color indicates a different functional group. Trial 2.

**Figure 2 animals-10-02368-f002:**
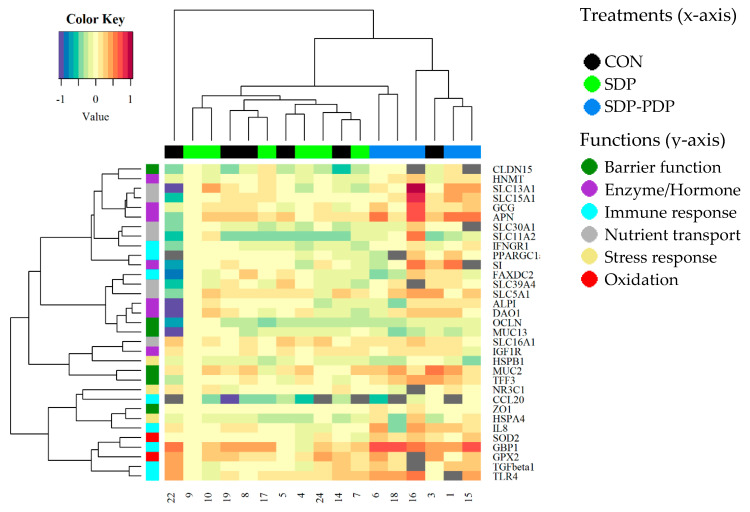
Heatmap representing the gene expression of each sample from jejunum in each gene in trial 2. CON: control diet; SDP: diet with 3% spray-dried plasma inclusion; SDP-PDP: diet supplemented with a combination of 1% spray-dried plasma and 2% porcine digestible peptides. Genes are organized in rows and samples in columns. Samples are labeled with different colors representing every treatment and genes depending on its functional group.

**Table 1 animals-10-02368-t001:** Composition of the experimental diets used on trial 1, % as fed basis.

	Experimental Diets ^1^		
Item	CON	SDP	PDP
Ingredient, %			
Maize	36.9	38.7	38.8
Wheat	16.0	16.0	16.0
Extruded Soybeans	15.0	11.4	11.2
Barley	13.0	13.0	13.0
Soybean meal 44% crude protein (CP)	6.8	6.8	6.8
Soybean protein concentrate 56% CP	5.6	5.6	5.6
Sweet milk whey	2.5	2.5	2.5
Spray-dried plasma 80% CP	-	2.0	-
Porcine digestible peptides 62% CP	-	-	2.0
Mono calcium phosphate	1.34	1.37	1.30
Calcium carbonate	0.62	0.64	0.67
L-Lysine HCl	0.55	0.49	0.52
Salt	0.53	0.40	0.16
Vitamin-Mineral premix ^2^	0.40	0.40	0.40
DL-Methionine	0.27	0.23	0.57
L-Threonine	0.25	0.21	0.23
L-Valine	0.15	0.11	0.18
L-Tryptophan	0.09	0.08	0.10
Calculated composition			
Net Energy (NE), kcal/kg	2470	2470	2470
Dry Matter, %	89.1	89.0	89.1
Ash, %	5.4	5.3	5.4
Crude Protein, %	19.5	19.7	19.5
Calcium, %	0.650	0.655	0.650
Total P, %	0.678	0.671	0.671
Digestible amino acids			
Lys, %	1.280	1.280	1.280
Met, %	0.509	0.470	0.829
Cys, %	0.222	0.262	0.213
Met+Cys, %	0.768	0.768	1.078
Thr, %	0.832	0.832	0.832
Trp, %	0.282	0.282	0.282
Val, %	0.896	0.896	0.934
Analyzed composition			
Dry Matter, %	89.1	88.8	88.8
Ether Extract, %	5.1	4.2	4.3
Neutral Detergent Fiber, %	10.5	10.2	10.0
Crude Protein, %	17.9	18.2	18.4

^1^ Experimental diets: CON: control diet; SDP: diet with 2% spray-dried plasma inclusion; PDP: diet with 2% porcine digestible peptides inclusion. ^2^ Supplied the following per kg of diet: 7000 IU of vitamin A (acetate); 500 IU of vitamin D3 (cholecalciferol); 250 IU of vitamin D (25-hydroxicholecalciferol); 45 mg of vitamin E; 1 mg of vitamin K3; 1.5 mg of vitamin B1; 3.5 mg of vitamin B2; 1.75 mg of vitamin B6; 0.03 mg of vitamin B12; 8.5 mg of D-pantothenic acid; 22.5 mg of niacin; 0.1 mg of biotin; 0.75 mg of folacin; 20 mg of Fe (chelate of amino acids); 2.5 mg of Cu (sulphate); 7.5 mg of Cu (chelate of glycine); 0.05 mg of Co (sulphate); 40 mg of Zn (oxide); 12.5 mg Zn (chelate of amino acids); 12.5 mg of Mn (oxide); 7.5 of Mn (chelate of glycine); 0.35 mg of I, 0.5 of Se (organic); and 0.1 mg of Se (inorganic).

**Table 2 animals-10-02368-t002:** Composition of the experimental diets used on trial 2 (%, as fed basis).

	Experimental Diets ^1^			
Item	CON	SDP	SDP-PDP	Starter
Ingredients, %				
Maize	29.68	32.53	32.11	29.79
Soybean Meal 47% crude protein (CP)	25.82	20.6	21.34	21.34
Wheat	16	16	16	15
Barley	6.5	6.5	6.5	20
Dextrose	6.5	6.5	6.5	-
Sweet Milk Whey	6.5	6.5	6.5	-
Potato Protein	2.5	2.5	2.5	-
Lard	2.67	2.38	2.31	6.53
Di-calcium phosphate	1.72	1.77	1.67	1.56
Spray-dried plasma (80% CP)	-	3	1	-
Porcine digestible peptides 62% PB	-	-	2	-
Salt	0.47	0.27	0.03	0.48
Vitamin-Mineral Premix ^2^	0.40	0.40	0.40	0.40
L-Lysine HCL (78)	0.46	0.39	0.39	0.51
DL-Methionine	0.27	0.24	0.25	0.25
L-Threonine	0.21	0.16	0.17	0.24
Calcium Carbonate	0.11	0.11	0.17	0.55
L-Valine	0.11	0.09	0.08	0.14
L-Tryptophan	0.08	0.07	0.09	0.07
Calculated composition				
Net Energy (NE), kcal/kg	2470	2470	2470	2653
Ether Extract, %	4.69	4.43	4.39	8.61
Neutral Detergent Fiber, %	7.87	7.57	7.61	9.92
Crude Protein, %	20.51	20.50	20.50	18.50
Calcium, %	0.65	0.65	0.65	0.73
Total P, %	0.70	0.68	0.68	0.65
Digestible amino acids				
Lys, %	1.350	1.350	1.350	1.230
Met, %	0.545	0.504	0.539	0.478
Met + Cys, %	0.810	0.810	0.810	0.720
Thr, %	0.878	0.878	0.878	0.780
Trp, %	0.297	0.297	0.297	0.264
Analyzed composition				
Dry Matter, %	90.3	92.3	91.8	89.3
Ether Extract, %	4.6	4.3	4.1	8.4
Neutral Detergent Fiber, %	9.7	7.7	8.2	-
Crude Protein, %	19.7	18.7	19.6	17.7

^1^ Experimental diets: CON: control diet; SDP: diet with 3% spray-dried plasma inclusion; SDP-PDP: diet supplemented with a combination of 1% spray-dried plasma and 2% porcine digestible peptides. ^2^ Supplied the following per kg of diet: 7000 IU of vitamin A (acetate); 500 IU of vitamin D3 (cholecalciferol); 250 IU of vitamin D (25-hydroxicholecalciferol); 45 mg of vitamin E; 1 mg of vitamin K3; 1.5 mg of vitamin B1; 3.5 mg of vitamin B2; 1.75 mg of vitamin B6; 0.03 mg of vitamin B12; 8.5 mg of D-pantothenic acid; 22.5 mg of niacin; 0.1 mg of biotin; 0.75 mg of folacin; 20 mg of Fe (chelate of amino acids); 2.5 mg of Cu (sulphate); 7.5 mg of Cu (chelate of glycine); 0.05 mg of Co (sulphate); 40 mg of Zn (oxide); 12.5 mg Zn (chelate of amino acids); 12.5 mg of Mn (oxide); 7.5 of Mn (chelate of glycine); 0.35 mg of I, 0.5 of Se (organic); and 0.1 mg of Se (inorganic).

**Table 3 animals-10-02368-t003:** Effect of the experimental treatments on growth performance of piglets in trial 1.

	Experimental Diets ^3^				
Item ^2^	CON	SDP	PDP	SEM	*p*-Value ^1^
BW, day 0, g	7505	7517	7534	12.0	0.241
day 14, g	9438 ^b^	10,381 ^a^	9990 ^a^	138.9	0.001
ADFI 0–14 d, g/d	236 ^b^	295 ^a^	250 ^b^	11.3	0.002
ADG 0–14 d, g/d	138 ^b^	205 ^a^	175 ^a^	11.8	0.002
G:F 0–14 d	0.554 ^b^	0.695 ^a^	0.700 ^a^	0.0413	0.034

^1^*p*-values come from the ANOVA test. ^2^ BW: body weight; ADFI: average daily feed intake; ADG: average daily gain; G:F: feed efficiency. ^3^ Experimental diets: CON: control diet; SDP: diet with 3% spray-dried plasma inclusion; SDP-PDP: diet supplemented with a combination of 1% spray-dried plasma and 2% porcine digestible peptides. ^a,b^ Different letters in the same row indicate significant statistical mean differences in the two diets (Tukey’s test *p*-value < 0.05).

**Table 4 animals-10-02368-t004:** Effect of the experimental treatments on growth performance of piglets in trial 2.

	Experimental Diets ^3^				
Items ^2^	CON	SDP	SDP-PDP	SEM	*p*-Value ^1^
BW, day 0, g	5724	5720	5721	46.9	0.859
day 7, g	5956 ^b^	6344 ^a^	6254 ^ab^	88.3	0.023
day 14, g	7165 ^b^	7894 ^a^	7871 ^a^	152.7	0.008
day 35, g	15,270	16,252	15,960	460.3	0.335
ADFI, 0–14 d, g	267	282	283	5.96	0.139
14–35 d, g	483	508	525	25.9	0.532
ADG, 0–14 d, g	103 ^b^	155 ^a^	153 ^a^	10.7	0.007
14–35 d, g	364	398	382	22.0	0.571
G:F, 0–14 d	0.380 ^b^	0.548 ^a^	0.540 ^a^	0.0383	0.046
14–35 d	0.761	0.782	0.732	0.0501	0.840

^1^*p*-values come from the ANOVA test. ^2^ BW: body weight; ADFI: average daily feed intake; ADG: average daily gain; G:F: feed efficiency. ^3^ Experimental diets: CON: control diet; SDP: diet with 3% spray-dried plasma inclusion; SDP-PDP: diet supplemented with a combination of 1% spray-dried plasma and 2% porcine digestible peptides. ^a,b^ Different letters in the same row indicate significant statistical mean differences in the two diets (Tukey’s test *p*-value < 0.05).

**Table 5 animals-10-02368-t005:** Effect of the experimental treatments on relative gene expression on jejunum mucosa after pre-starter period. Gene expression values are presented as ratios of cycle relative threshold value for each gene normalized to that of the reference sample. *p*-Values come from the ANOVA test and FDR is the false discovery rate.

Function	Genes ^1^	Experimental Diets ^2^			Contrast Statistic	*p*-Value	*q*-Value (FDR)
		CON	SDP	SDP-PDP			
Barrier function	CLDN15	0.51 ^b^	0.81 ^ab^	1.05 ^a^	4.984	0.027	0.139
	TFF3	1.15 ^b^	1.07 ^ab^	1.83 ^a^	4.290	0.035	0.145
Immune response	TLR4	1.54 ^b^	1.31 ^b^	2.86 ^a^	8.052	0.005	0.059
	GBP1	2.40 ^ab^	1.80 ^b^	4.20 ^a^	5.711	0.015	0.127
	IFNGR1	0.69 ^b^	0.74 ^ab^	1.10 ^a^	4.661	0.028	0.139
Nutrient transport	SLC11A2/DMT1	0.37 ^b^	0.71 ^ab^	1.24 ^a^	3.862	0.046	0.169
Enzyme/Hormone	HNMT	0.86 ^b^	0.91 ^b^	1.56 ^a^	14.111	4 × 10^−4^	0.015
	APN	1.59 ^b^	1.40 ^b^	3.40 ^a^	4.577	0.030	0.139
Oxidation	SOD2	1.04 ^b^	0.95 ^b^	1.71 ^a^	11.030	0.001	0.022

^1^ Genes: CLDN15: claudin 15; TFF3: trefoil factor 3; SLC11A2/DMT1: solute carrier family 11 member 2/divalent metal transporter 1 (DMT1); TLR4: Toll like receptor 4; GBP1: guanylate binding protein; IFNGR1: interferon gamma receptor 1; HNMT: histamine N-methyltransferase; APN: alanyl aminopeptidase; SOD2: superoxide dismutase 2. ^2^ Experimental diets: CON: control diet; SDP: diet with 3% spray-dried plasma inclusion; SDP-PDP: diet supplemented with a combination of 1% spray-dried plasma and 2% porcine digestible peptides (2%). ^a,b^ Different letters in the same row indicate significant statistical mean differences in the two diets (Tukey’s test *p*-value < 0.05).

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
