# Peer review of "Porcine Digestible Peptides (PDP) in Weanling Diets Regulates the Expression of Genes Involved in Gut Barrier Function, Immune Response and Nutrient Transport in Nursery Pigs"

_animals, 2020, doi:10.3390/ani10122368_

Round 1

Reviewer 1 Report

The paper from Gonzalez-Sole and co-authors describes a study investigating the use of porcine digestible peptides in weaner pig diets and the impacts on gene expression in the GIT of genes related to gut function and gut health. Overall the manuscript is well written, and the topic has been well researched. The writing is in clear English and the experimental design has been well described. The introduction, materials and methods, and discussion sections are well written and structured, and give a clear and concise overview of the existing literature and the experimental design of the current experiment.

Most of my comments suggest minor changes in relation to wording and formatting (please see attachment). Otherwise, well done to these authors for a very well presented manuscript.

Author Response

The response letter to the comments of reviewer 1 are correspondingly attached to avoid format incompatibilities as they are in table format.

Reviewer 2 Report

The paper addresses a practical question; the value of plasma and PDP, and it even attempts to dive into their mode of action. Especially the latter work is valuable. Studies seem to be performed well, and the paper is generally informative and well written. Nevertheless, I stumble upon an inconsistency that bothers me; in the introduction, the focus is the extra-nutritional properties of the test products. However, the discussion starts with the negative effects of soy products, which were substituted by the test products. Indeed, these two are unfortunately intertwined in this study making it very hard to include if one was more positive than the other one was negative (I would have substituted a more neutral protein source). In my eyes, this dilemma should be incorporated both in the introduction and in the discussion and conclusion; the substitution of soy products by plasma and/or PDP resulted in…. (and explain why).

For the genome work, is it possible to speculate a bit more how it all fits together? 

Another thing that I miss is that in my mind PDP is very high in salt. If this is indeed correct, then please note this in the paper and describe levels and possible consequences thereof.

Specific comments:

Missing: Nutrient profile of PDP and SDP

Table 2: Maize is used in the first table; please delete Corn

Vitamin-mineral premix: report consistently to 2 decimals (0.40)

149: diets for trial 1 were not analyzed?

208: PCA is very much a descriptive technique; why was no PLS(-DA) analysis used?

Table 3 and 4: performance (ADG, ADFI) are reported with 4 decimals, and G/F with 2. Any logic behind this? Personally, I prefer 3 for all performance parameters.

Table 5: are the superscripts right for TFF?

276: pity that the function of these genes is hidden as a footnote to a table

Fig 1: legend for functions is not the clearest; would be nicer if it showed lines instead of bullets, and if the colors were more distinct (orange x2)

Ileum; dangerous to do stats with only 3 control samples…

Fig 2: I realize that the genomics folks love these graphs, but I find them everything but useful. As a minimum, I would group animals on the same treatment together; do the branches on the top have any value for this paper?

The legend for function and the color key are too similar. Please clarify what links to what (e.g., clarify that the left column is ‘function’).

335: by highlighting the problems of soy you agree with me that the conclusion of this trial could just as well relate to the removal of soy than by the introduction of animal protein into the diet? That would require a completely different introduction…

348: demonstrate that including.. No; the trial demonstrates that replacing soy with animal protein is positive!

352: all these --> both

353: clear positive control: what do you mean?

368-370: or to the removal of soy.

391, 393: two 1-sentence paragraphs

470: bioactive peptides that work orally are still rare

Author Response

The corresponding response to comments of Reviewer 2 are enclosed in the attached file in order to avoid compatibility format problems due to the table format.

Reviewer 3 Report

The article describes two trials with pigs in their post-weaning period, that receive PDP and/or SDP as substitution of SBM. The authors show significant effects on growth performance parameters. They also claim to find significantly expressed genes, however this is a major concern for me in the reporting. IN fact there are only 2 genes significantly expressed in jejunum, namely HNMT and SOD2, because the have a FDR<0.05 (which is often the threshold used in gene expression analysis). All other reported genes are therefore per definition not significant.

The trials have different charateristics to them, such as different weaning dates, why is so. The authors do not explain why they have changed this over the two trials, but could have an effect on the intestinal functioning.

I am not an expert in diet composition, however I did notice that the authors state that the SBM was replaced by SDP/PDP, however the corn percentage included was also slightly higher. Do they expect some differences of biofunctional properties in the gastrointestinal tract as well? please elaborate.

The effect on the growth performance parameters was mainly shown during or just after the experimental period of the treatments. In the second trial there even looks like to be a compensatory growth of the controls, i.e. there is no significant effect in BW on d35. And for the other parameters, ADFI, ADG, G:F the effects are only in the period day 0-14. 

In the discussion section, the authors state the following (line 361-362): "Consumption of feed containing the hydrolysates, PDP or SDP-PDP, enhanced the productive performance of piglets compared to their corresponding control group", I would add here that this only holds for the starter phase, in other words the authors should more nuance their findings.

In line 367, the relation between gut health and villus height is made, I personally do not believe that is such a clear one-on-one relation as is depicted here. This needs more nuance.

Line 371-373, repetition of lines 366-367.

In line 373-374, it is said that this is the first study with gene expression and PDP on intestinal functionality, although this is true, I believe when measuring only up to 60 genes this only reflects part of intestinal functionality. 

Line 480-482, is overselling it in my opinion, underlying mechanisms are not really found or discussed based on these results.

Overall, the claims made are not substantiated by the shown results.    

Author Response

REVIEWER 3

The article describes two trials with pigs in their post-weaning period, that receive PDP and/or SDP as substitution of SBM. The authors show significant effects on growth performance parameters. They also claim to find significantly expressed genes, however this is a major concern for me in the reporting. IN fact there are only 2 genes significantly expressed in jejunum, namely HNMT and SOD2, because the have a FDR<0.05 (which is often the threshold used in gene expression analysis). All other reported genes are therefore per definition not significant.

Authors: Following the reviewer’s advice we have edited the whole manuscript, both eliminating the results from the ileum, in which all changes were not significant (FDR, Q > 0.05) but also differentiating the observed trends (P < 0.05, Q < 0.2) and the significant differences P < 0.05, Q < 0.05). This implementation has implied changes in the MM section, the results but also has helped us to shorten and better focus the discussion.

The trials have different characteristics to them, such as different weaning dates, why is so. The authors do not explain why they have changed this over the two trials, but could have an effect on the intestinal functioning.

  • Authors: Thank you for your comment. The two trials were performed in different farms and the differences in the experimental design between them (i.e. weaning date) were not our decision but a consequence of the disparities in the logistics of each farm. We agree that it is highly possible that this has influenced the intestinal functioning of the animals; for instance, in the second trial animals were at a more immature stage (21d of life) and probably they were more challenged than the ones in the first trial. However, results from the two experiments are very similar, so we could deduce that the effect of PDP and SDP is present in both situations. We have tried to make some comment to this difference on the trials design in lines 119 to 121.

I am not an expert in diet composition, however I did notice that the authors state that the SBM was replaced by SDP/PDP, however the corn percentage included was also slightly higher. Do they expect some differences of biofunctional properties in the gastrointestinal tract as well? please elaborate.

  • Authors: Thank you for your comment. The higher inclusion of corn in the SDP and PDP diets became necessary after feed formulation to maintain the same levels of nutrients in all diets. The addition of SDP and PDP compensate the reduction of SBM or extruded soybeans in terms of protein, but energy has to be covered by an increase on corn inclusion. At this respect, corn becomes a major ingredient in all these diets, without a functional role other than satisfying nutrients, such as energy and protein. From our experience, the required changes on corn, associated to feed formulation, are not expected to contribute significantly to the observed results.

The effect on the growth performance parameters was mainly shown during or just after the experimental period of the treatments. In the second trial, there even looks like to be a compensatory growth of the controls, i.e. there is no significant effect in BW on d35. And for the other parameters, ADFI, ADG, G:F the effects are only in the period day 0-14. In the discussion section, the authors state the following (line 361-362): "Consumption of feed containing the hydrolysates, PDP or SDP-PDP, enhanced the productive performance of piglets compared to their corresponding control group", I would add here that this only holds for the starter phase, in other words, the authors should more nuance their findings.

  • Authors: Thank you for your comment. We agree this statement has to be nuanced. It has been clarified that the increase of the growth performance only happended at the end of the pre-starter phase, every time it has appeared in the discussion. (Lines 339, 348 and 440)

In line 367, the relation between gut health and villus height is made, I personally do not believe that is such a clear one-on-one relation as is depicted here. This needs more nuance.

  • Authors: We understand the suggestion. We have done changes nuancing our affirmations, defining the villus height as an indicator of good nutrient uptake instead of gut health. (Lines 353-355)

Line 371-373, repetition of lines 366-367.

  • Authors: One of the sentences has been removed. Line 372.

In line 373-374 (after correction lines 358-359), it is said that this is the first study with gene expression and PDP on intestinal functionality, although this is true, I believe when measuring only up to 60 genes this only reflects part of intestinal functionality.  

  • Authors: We agree with your comment. It is true that this set of genes is not enough to monitor extensively the intestinal functionality of the pig, but in this research, our goal was to give a first overview of the effects that PDP could have in different specific areas of the intestinal function (barrier function, immune system, nutrient transport…). We changed some statements of the manuscript that were too categorical by explaining that we intended to provide some inputs about the effects of PDP on intestinal function. (Lines 33-34, 45-47, 75, 70, 358, 435-437 and 445-447).

Line 480-482, is overselling it in my opinion, underlying mechanisms are not really found or discussed based on these results.

  • Authors: We understand you comment. Attending to the fact that most of the differences found in the gene expression cannot be considered significant due to the FDR, before referring to the underlying mechanisms, more research should corroborate that PDP can produce an effect over the intestinal function. We have introduced a statement about this in the manuscript (lines 435-437 and 445-447). Referring to the bioactive peptides, we had information that other researchers were working on that line and they had found positive results. However, this research is still in progress and there are no available publications nor information related to it because it is under the effects of a patent and nothing can be published until 18 months from now. As we do not have any reference to support that PDP might contain bioactive peptides we have decided to delete this claim.

Overall, the claims made are not substantiated by the shown results.

  • Authors: Thank you for helping us improve our work. We hope that our modifications have successfully responded to your comments and now the manuscript is acceptable for publishing.

Round 2

Reviewer 3 Report

Please check the English of the newly written parts, e.g. "doesn't" should be "does not"

Author Response

The changes have been done with the track changes function of Microsoft word and are detailed below:

146/147: a comma and a verb in the next sentence have been added.

273: “Showed also” has been substituted by “also showed”.

307: “On the other hand” has been substituted by “In addition”

320: The preposition “by” has been removed from the sentence.

321: “Doesn’t” has been changed to “does not”.

We hope that our modifications have been enough to